# Striving for data-model efficiency:
# Identifying data externalities on group performance

**Esther Rolf**[*]
Harvard University

**Ben Packer**
Google Research

**Alex Beutel**
Google Research

**Fernando Diaz**
Google Research

## Abstract

Building trustworthy, effective, and responsible machine learning systems hinges on understanding how differences in training data and modeling decisions interact to impact predictive performance. In this work, we seek to better understand how we might characterize, detect, and design for data-model synergies. We focus on a particular type of data-model inefficiency, in which adding training data from some sources can actually lower performance evaluated on key sub-groups of the population, a phenomenon we refer to as *negative data externalities on group performance*. Such externalities can arise in standard learning settings and can manifest differently depending on conditions between training set size and model size. Data externalities directly imply a lower bound on feasible model improvements, yet improving models efficiently requires understanding the underlying data-model tensions. From a broader perspective, our results indicate that data-efficiency is a key component of both accurate and trustworthy machine learning.

## 1   Introduction

Although key aspects of trustworthiness and responsibility in machine learning are often framed from an algorithmic perspective, we explore an alternative framing that focuses on how our chosen modeling and training procedures perform under different data (collection) regimes. We refer to the guiding goal of using the available training data to achieve the best possible performance for all target populations as "data efficiency." While this has clear alignment with accuracy maximization, data minimization, and fairness, it is not always clear how to test or design for data-efficiency generally.

We focus on a specific type of data *inefficiency*, in which adding training data from certain data sources can actually decrease performance on key groups of the target population. We call this phenomenon *negative data externalities on group performance* (defined formally in definition 1). While recent works have examined how the amount of training data from one group affects performance evaluated on that same group [7, 11, 19], studying *between-group* trends can evidence tensions between different aspects of model performance that might otherwise go unnoticed.

In this work, we formalize data externalities and their immediate cost to model performance (§2,§3). We describe mechanisms by which negative data externalities can arise and detail why understanding these mechanisms matters for designing effective model improvements (§4). Our experiments show how data externalities shift with conditions on sample size and model capacity (§5). Altogether, our results suggest that data externalities may be suprisingly pervasive in real-world settings, yet avoidable by designing training procedures to promote data-efficiency. Our findings have implications to fairness, privacy, participation, and transparency in machine learning (§6).

---

[*]Work done while interning at Google Research.

2022 Trustworthy and Socially Responsible Machine Learning (TSRML 2022) co-located with NeurIPS 2022.

**Related Work.**    Alongside a recognition of the role of training data in achieving high-performing models [1, 19, 20] has come an appreciation that dataset size does not imply requisite quality [4], representativeness [6], or diversity [3]. For example, concerns have been raised about the use of large, multi-source text datasets, including transparency of data provenance [10] contributed to by a "documentation debt" [3]. In this work, we study how training data from different sources can affect model performance, focusing on possible externalities to key subgroups of the target population.

Prior work has proposed scaling laws to describe how the total number of training samples and model size affect aggregate performance [2, 16]. Our study connects to a growing line of research on how varying the amount of training data from *different* sources, groups, or classes impacts model performance, including subgroup and average accuracies [13, 19], and statistical fairness [7, 11]. Scaling laws that have been proposed to model the impact of data from one dataset (e.g. [2, 16]) or multiple source groups (e.g. [7, 19]) on model performance typically implicitly encode that more data never worsens performance, and thus do not allow for data externalities of the type we study here. Data valuation techniques have been proposed to quantify the (possibly negative) value of individual data points [12, 18]. In a framework more similar to our own, [15] studies scenarios where leaving out entire classes from a training dataset can improve transfer learning performance.

Finally, we note that the general terms "data externalities" and "information externalities" have also been used to describe externalities on other factors, such as privacy [8] and data valuation [14].

## 2   Setting and notation

We assume that each training instance is collected from exactly one of $g \in \mathcal{G}_{\text{source}}$ distinct source groups. Each evaluation instance can be associated with any subset of predefined evaluation groups $g \in \mathcal{G}_{\text{eval}}$, which may be intersecting. In this work, we assume that groups are known at training and evaluation time. We will often make the simplifying assumption that $\mathcal{G}_{\text{eval}} = \mathcal{G}_{\text{source}}$ referring to the set of groups simply as $\mathcal{G}$.

We assume that one can sample instances $(x, y) \sim \mathcal{D}_g$ for each group in $g \in \mathcal{G}_{\text{source}} \cup \mathcal{G}_{\text{eval}}$, where $\mathcal{D}_g$ denotes the data distribution corresponding to group $g$. We say that a model $f(\cdot)$ results in group risks, or expected losses of $\mathcal{R}(g) = \mathbb{E}_{(x,y) \sim \mathcal{D}_g}[\ell(f(x), y)]$ defined with respect to to loss function $\ell$.

**Per-group risks as random variables parameterized by training dataset composition.**    Following [19], we consider the learned model $f_\theta$ as a random function parameterized by both the model parameters $\theta$ and allocations governing sample sizes from each source. Denote the (random) training dataset $\mathcal{S}$ as the union of the $|\mathcal{G}_{\text{source}}|$ sets each comprising samples from one training source:

$$\mathcal{S}(n_1, \ldots, n_{|\mathcal{G}_{\text{source}}|}) = \bigcup_{g \in \mathcal{G}_{\text{source}}} \{(x_i, y_i) \sim_{i.i.d.} \mathcal{D}_g\}_{i=1}^{n_g} \tag{1}$$

where sample sizes are determined by the total dataset size $n = \sum_{g \in \mathcal{G}_{\text{source}}} n_g$ and *allocations* $n_g/n$. With these definitions in place, we can define the expected risk of a training procedure as a function of the composition of the training set. For example, for a standard training procedure that selects a model from class $\mathcal{F}_\Theta = \{f_\theta\}_{\theta \in \Theta}$ to minimize loss $\ell_{\text{train}}$:

$$\mathbb{E}[\mathcal{R}(g_{\text{eval}}); \{n_g\}_{g \in \mathcal{G}_{\text{source}}}; \mathcal{F}_\Theta, \ell_{\text{train}}] = \mathbb{E}_{\mathcal{S}(\{n_g\})} \left[ \mathbb{E}_{(x,y) \sim \mathcal{D}_{g_{\text{eval}}}} \left[ \ell(f_\theta(x), y) | \theta = \theta^*_{\ell_{\text{train}}, \mathcal{S}} \right] \right] \tag{2}$$

where $\theta^*_{\ell_{\text{train}}, \mathcal{S}} = \arg\min_{\theta \in \Theta} \ell_{\text{train}}(f_\theta, \mathcal{S})$. Equation (2) gives us a framework by which to describe performance of our learning procedure across evaluation groups as sensitive to many different scenarios and choices, including the training source sample sizes $n_g$ and model class complexity.

## 3   Data externalities and their costs

We now use the framework of §2 to characterize our main phenomenon of study: when adding training data from a particular source group decreases performance for an evaluation group.

**Definition 1 (negative data externality on group risk).** For fixed training procedure described by model class $\mathcal{F}$ and loss objective $\ell_{\text{train}}$, a negative data externality on group risk (w.r.t. groups $\mathcal{G}_{\text{eval}}$)

occurs when the expected risk for some evaluation group can *increase* as a result of *adding* randomly sampled training data:

$$\exists\, g_{\text{eval}} \in \mathcal{G}_{\text{eval}},\ \{n'_g \leq n_g\}_{g \in \mathcal{G}_{\text{source}}}: \tag{3}$$

$$\mathbb{E}[\mathcal{R}(g_{\text{eval}}); \{n'_g\}_{g \in \mathcal{G}_{\text{source}}}; \mathcal{F}, \ell_{\text{train}}] < \mathbb{E}[\mathcal{R}(g_{\text{eval}}); \{n_g\}_{g \in \mathcal{G}_{\text{source}}}; \mathcal{F}, \ell_{\text{train}}].$$

We can interpret eq. (3) across all possible allocations that could be collected, or with respect to an existing dataset with total size $n = \sum_{g \in \mathcal{G}_{\text{source}}} n_g$, where $\mathcal{D}_g$ denote empirical distributions and sampling is done uniformly at random. In the second case, a data externality exposes an implicit "cost" to some evaluation group, formalized as a room for improvement, $\Delta$, in the following claim.

**Claim 1** (Data externalities lower bound room for model improvement). For a training dataset with $\{n_g\}_{g \in \mathcal{G}_{\text{source}}}$, for fixed training procedure with model class $\mathcal{F}$ and training loss $\ell_{\text{train}}$, the maximum magnitude of data externality $\Delta_{g_{\text{eval}}}$ on group $g_{\text{eval}}$,

$$\Delta_{g_{\text{eval}}} := \max_{n'_g \leq n_g \forall g} \left[ \mathbb{E}[\mathcal{R}(g_{\text{eval}}); \{n_g\}_{g \in \mathcal{G}_{\text{source}}}; \mathcal{F}, \ell_{\text{train}}] - \mathbb{E}[\mathcal{R}(g_{\text{eval}}); \{n'_g\}_{g \in \mathcal{G}_{\text{source}}}; \mathcal{F}, \ell_{\text{train}}] \right], \tag{4}$$

is a lower bound on the best possible improvement in expected risk for group $g_{\text{eval}}$ that can be achieved using this dataset *without raising the expected risk for groups disjoint from $g_{eval}$*.

*Proof.* We can construct an alternative training procedure that first subsets the training data uniformly at random from each source group according to the $n'_g$ that maximize the expression in eq. (4). Since groups are assumed to be known, we can selectively apply this new model to instances from $g_{\text{eval}}$ and use the original model for all other instances. This split model lowers expected risk by $\Delta_{g_{\text{eval}}}$ for $g_{\text{eval}}$ and does not alter expected risk for instances not in $g_{\text{eval}}$. As this procedure only optimizes over the training data sub-sampling, it is a *lower bound* for the possible performance improvements. $\square$

Claim 1 highlights that identifying data externalities can improve the model for groups on which the model under-performs, without any negative consequence to expected risk evaluated on disjoint evaluation groups. However, data externalities also tell us something more subtle about the compatibility of our model with the underlying structures in our data. In the next section, we investigate possible causes of data externalities, and what they mean for improving model performance.

## 4 When do negative data externalities arise?

An intuitive setting where data externalities can arise is when the complexity of model class $\mathcal{F}_\Theta$ is constrained or mis-specified so that the optimal parameters differ per group, i.e. $\theta_g^* \neq \theta_{g'}^*$, where $\theta_g^* := \arg\min_{\theta \in \Theta} \mathbb{E}_{(x,y) \sim \mathcal{D}_g}[\ell(f_\theta(x), y)]$. Here we detail such a setting, as well as another example in which data externalities arise even when the optimal model is the same for all groups ($\theta_g^* = \theta_{g'}^*$).

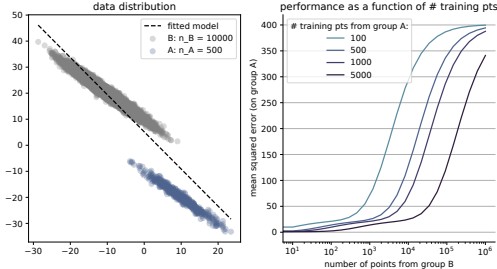 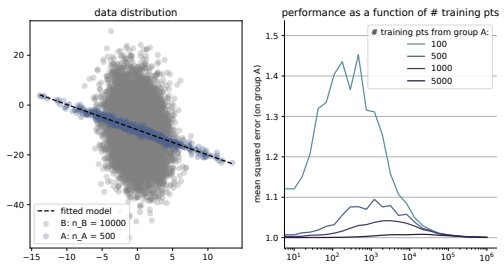

(a) **Insufficient/mis-allocated model capacity**: The true intercept of optimal model differs between groups but model class requires a fixed intercept. This results in data externalities on one group as the number of training samples from the other group increases.

(b) **Different group data distributions**. The true model is shared between groups, but the distributions of features and observation noise differs. Adding small $n_B$ can detract from performance on group A due to increased noise in the training data.

Figure 1: Two illustrative examples based on the data-generating distribution described in §4.

Our examples use an illustrative model where we assume there exists a true affine relationship for each group, with shared linear weights but different intercepts (see §A.3 for exact parameters):

$$y_g = w \cdot x_g + b_A \cdot \mathbb{I}[g = A] + b_B \cdot \mathbb{I}[g = B] + \epsilon_g \quad g \in \{A, B\} = \mathcal{G} \ ,$$

but the model class is the set of affine models shared between groups ($f_\theta(x, g) = \theta_1 x + \theta_2$). In the first example, we vary the intercept of the true model between the groups, as well as the mean of the feature distribution (fig. 1a, left). The discrepancy between the true model and the model class results in data externalities with magnitude ($\Delta$ in eq. (4)) increasing monotonically as the number of samples from the other group increases (positive slopes in fig. 1a, right).

In the second example, the same model parameters apply to both groups, but we vary the spread of the feature distribution and scale of observation noise $\epsilon_g$ between groups, effectively decreasing the signal-to-noise ratio for group $B$ relative to group $A$ (fig. 1b, left). This results in data externalities evaluated on group $A$ for small to mid-range number of samples from group $B$, which dissipate with larger $n_B$ (fig. 1b, right). What constitutes "small to mid-range" values of $n_B$ is relative to $n_A$: the magnitude of the negative data externalities decrease with sufficient samples from group $A$.

In these two examples, negative data externalities arise for different reasons and **the modeling intervention that would best address the data externalities depends on the cause of the tension**. In the first example, allowing for a more complex model class which fits a different intercept term for each group (expanding model capacity in a targeted way) will alleviate data externalities for any $\{n_A, n_B\}$. In the second example, removing negative data externalities by splitting the model by group would eliminate the positive externalities of adding samples from group B when $n_B$ is large ($\geq 1e5$ in fig. 1b). A more appropriate strategy for the second example would reweight instances according to their source group in when computing and optimizing the training loss.

While data externalities signal a clear opportunity to improve performance (claim 1), the examples above highlight that best way to make model improvements will depend on the setting. Data externalities could thus be considered as a symptom indicating sub-optimal data-efficiency of a given modeling procedure. Remedying the exposed tensions in an effective manner will require understanding the underlying mechanisms giving rise to the observed data externalities.

## 5 Data externalities with real data

Experiments in this section expose negative data externalities with respect to the empirical distributions defined by two different real-world datasets (see §A.1 for more details on datasets):

The **Goodreads datasets** [22] contain book reviews and ratings of books from different genres. We collect data for two genres – history/biography (*history*) and fantasy/paranormal (*fantasy*) – which comprise the two groups $g$ in our setup. Similar to in [19], the binary prediction task is to discern whether a book review corresponds to a 5 star rating (1) or less (0), given the text of the review.

The **CivilComments dataset with identity information** [5] contains online comments with human-annotated labels corresponding to whether the comment is considered toxic and whether the comment is targeted at a specific group $g$. We focus on the four largest identity groups present in the dataset: *female, male, Christian, and Muslim* (groups are determined as binary labels if the annotator average is at least $0.5$ for that identity group, similar to toxicity labels).

Beyond evidencing that negative data externalities can manifest in real-data settings, we design experiments to understand *when* they manifest, in light of results from §4. We examine how data externalities arise under different conditions on total sample size (§5.1) and model capacity (§5.2).

To identify data externalities with existing datasets, we sub-sample the available training data from each source group $g$ uniformly at random with different allocations defined by $\{n_g\}_{g \in \mathcal{G}}$. To estimate eq. (2) for each group, we measure the per-group performance of the resulting models on fixed evaluation sets. We report per-group area under the reciever operating characteristic (AUROC) as a metric that is insensitive to class imbalances (see §A.2). We report variability across multiple random draws of the training set for each allocation $\{n_g\}_{g \in \mathcal{G}}$.

### 5.1 Data externalities and dataset size (book rating prediction)

We first examine how data externalities manifest across different possible sample sizes and ratios between groups, using the rating prediction task described above. To compare performance across

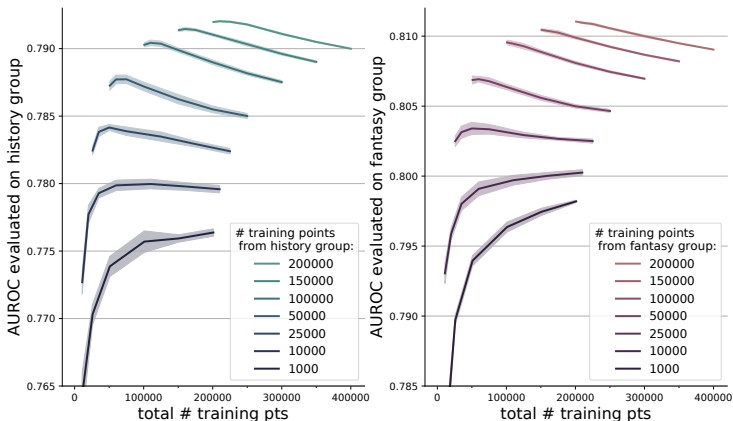

Figure 2: **Data externalities manifest when there is sufficient training data from the group being evaluated.** Each curve fixes the number of training points corresponding to the evaluation group. From left to right, training data is added randomly from the other genre. Solid lines show average per-group performance; shaded regions show 2 standard errors above and below the mean over 10 trials. Negative slopes diagnose data externalities measured with respect to per-group AUROC.

many subsets of training data, we chose a model that is fast to train. Following [7, 19], we use a linear regression model with $\ell_1$ penalty (lasso), trained on 1000-dimensional tf-idf vector embeddings of the review texts. The $\ell_1$ penalty is chosen via a cross-validation search for each subset.

Figure 2 shows that **negative data externalities can arise in real-data settings**, evidenced by the decrease in AUROC evaluated one group as data from another group is added to the training set (left to right on horizontal axis). As discussed in claim 1, these externalities provide obvious modeling interventions to increase per-group performance. For each panel (evaluation group) in fig. 2, we could subset the full training data to the allocation that maximizes AUROC along the vertical axis, resulting in two models trained with different training subsets, and as a result obtaining greater performance on each group (compared to performance with the full training set, $n = 400,000$).

The magnitude of the externality (measured from peak of each curve to rightmost point) is small, as expected from previous work that assumes between-group trends have a negligible effect on per-group performance as a function of training set allocations [7, 19]. Nonetheless, the existence of data externalities suggests that in some contexts, more nuanced scaling laws would be appropriate for describing model performance across data allocations.

The curves in fig. 2 are not all monotonic, meaning there is not an "all or nothing" answer to whether merging or splitting training data from multiple source groups optimizes model performance: sometimes, the best performance for group A is achieved by adding a moderate amount of training samples from group B. In fact, negative data externalities tend to manifest only once a certain number of points from the group in question are present in the training set (lighter-colored curves).

Drawing on our understanding from §4, we hypothesize that as the total number of training points increases, the training data "saturates" the model class, in the sense that the variance reduction due to additional points from group B is not worth the bias away from the optimal model parameters with respect to group A's data distribution. The exact saturation point would depend on the distribution of features and labels in each group and the distance between the group-optimal model parameters, the latter depending on the capacity of the model class. To examine this further, our next set of experiments examines how data externalities manifest with models of different capacity.

## 5.2 Data externalities and model size (toxicity classification)

We now examine how data externalities can differ for models of different sizes, using the CivilComments with identities dataset described above. We fine-tune pre-trained miniature BERT models of different model capacity from [21]. Model architectures are determined by the number of transformer layers $L$ and hidden embeddings size $H$ with corresponding number of attention heads. For each

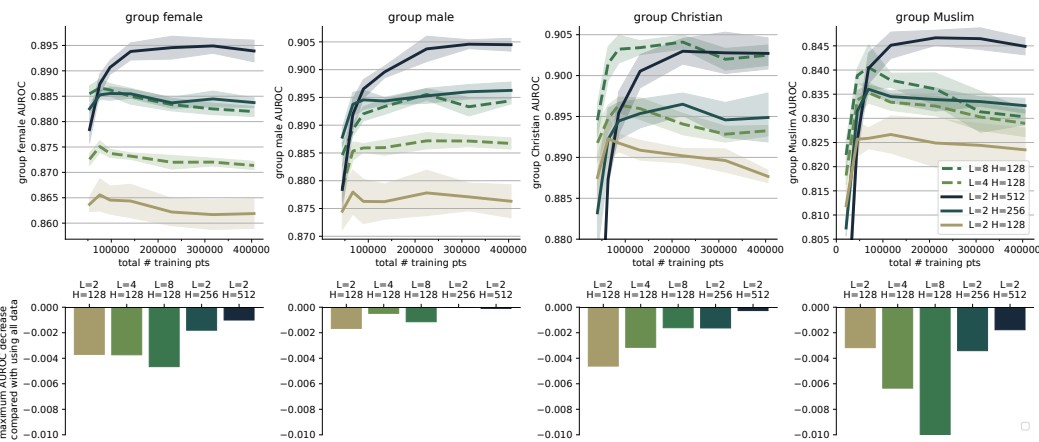

Figure 3: **Increasing model capacity has a complicated effect on magnitude of data externalities.** Performance and data externalities are measured across groups in the CivilComments dataset [5] for different miniature BERT models [21] with size determined by $L, H$. Top: group AUROCs for different training configurations and model architectures; shaded areas denote 2 standard errors above and below the mean across 5 trials. Bottom: the magnitude of data externalities from the top row.

fine-tuning run we use the Adam optimizer with learning rate 0.0001 and weight decay of 0.01 and train for 100 epochs with batchsize 64 and 500 gradient steps per epoch (see §A.2).

The leftmost points in each top panel of fig. 3 start with the maximum number of training points from the given group and increase the number of training points by adding data at random from the rest of the training set. The different hues in fig. 3 rank the models in terms of number of overall parameters (see table 1 in [21]). Negative slopes in the top row of fig. 3 (and corresponding negative values in the bottom row of fig. 3) evidence that data externalities arise in this data context and prediction task.

While increasing width or depth of the miniature BERT models (generally moving left to right on the bottom panels) *can* decrease the exhibited data externalities on per-group AUROC, **increasing model complexity does not necessarily mitigate negative data externalities on group performance, and in some cases can exacerbate them**. Adding additional layers to the model can increase the magnitude of data externalities evidenced (first three bars of groups *female, Muslim* in fig. 3, bottom), even though models with more layers tend to have higher overall performance. While this phenomenon tends to be more stark for models of increasing depth, we caution interpretation of the relative merits of adding model capacity with either depth or width without further analysis.

Taken together, the results in §5.1 and §5.2 evidence that data externalities manifest in different real-world data contexts and shed light on when and why they might manifest. Results in fig. 2 suggest that data externalities arise for one group primarily when there is enough representation in the training data from that group to "saturate" the model. Results in fig. 3 highlight that this point of saturation may depend on the complexity of the model parameterization among other factors. Future work could leverage these findings to build a stronger understanding of how model capacity might be tailored to jointly increase performance while promoting data efficiency under different conditions.

# 6  Discussion and open questions

We have shown that data externalities, a phenomenon in which adding more data from some input sources reduces performance on key evaluation groups, can occur in many machine learning settings. While this specific type of "data inefficiency" indicates room for model improvement, eq. (4) is likely to be a coarse lower bound for the possible improvements that could be made. Furthermore, the simple model modification described in the proof of claim 1 is only computationally reasonable when the number of evaluation groups is relatively small. Characterizing when and how data externalities can be (i) reliably identified for unknown evaluation groups or large number of groups and (ii) effectively mitigated within reasonable computational limits will be important future work.

We have focused on understanding how and when data externalities manifest across learning settings and training procedures. It would be interesting interesting to study data externalities and data-efficiency more generally as a principle by which to design algorithms from the outset. For example, a learning procedure guaranteeing no data externalities could enhance transparency regarding how input data affects model outputs, toward aligning of the goals of data minimization, participatory approaches, and fairness with traditional performance optimization in machine learning.

## Acknowledgements

The authors thank Jilin Chen and Pranjal Awasthi for providing feedback during the conceptualization, development, and writing of this work. We thank Yoni Halpern for feedback during the writing of this manuscript and Fernanda Viegas for feedback during figure development.

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

## A  Appendix

### A.1  Background and context for datasets

**Goodreads.**  The Goodreads datasets contain a corpus of book reviews across several genres, with numerical ratings and information about the book and reviewer [22]. This dataset and task have been used to study group effects across author genders and book genres in previous work [7, 19].

We follow a preprocessing approach similar to that in [19], where groups are genres. Specifically, we instantiate our experimental dataset with reviews from two genres: history/biography (*history*) and fantasy/paranormal (*fantasy*), excluding the few books with overlap between these genres. We take 250,000 review instances at random from the most popular 1000 books per genre, and split this data into group-balanced training set of 400,000 instances and validation/test set of 100,000 instances.

We consider binary labels of whether the rating accompanying each review text is a 5 star or less rating (originally measured on a 1-5 scale). The class distribution similar between groups, with an average label of 0.39 across history group training instances, and 0.41 across fantasy training instances. Review texts are embedded using tf-idf vectorization with 1000 features, and ignoring 'english' stopwords. The tf-idf vectors are computed with respect to the entire 500,000 instances.

**CivilComments (with identities) dataset for toxicity classification in text.** Classifying toxic comments in a corpus of text can be challenging due to different meanings or sentiments of words or phrases when they are used in reference to certain groups or topics [9]. The CivilComments dataset with identity information [5] contains online comments with human-annotated labels corresponding to whether the comment is considered toxic and whether the comment is targeted at a specific group.

As noted in [17], the test set for some identity groups can be very small. We focus on the four largest identity groups present in the dataset: female, male, Christian, and Muslim. Groups are determined as binary labels if the annotator average is at least $0.5$ for that identity group; binary toxicity labels are assigned similarly. The training sets contain 405,130 total instances, with 53,429 instances corresponding to the female group, 44,848 instances corresponding to the male group, 40,423 instances corresponding to the Christian group, and 21,006 instances corresponding to the Muslim group. The average toxicity rate (average binary label) differs across groups: in the training set the toxicity rates by group are: 14% (female), 15% (male), 11% (Christian), and 24% (Muslim).

To increase the number of evaluation samples across groups, we combine the validation and test sets in our analysis. Combining the validation and test sets means we don't have a separate validation set to tune model hyper-parameters. Should a separate validation set be necessary, one could use the pre-processed dataset from [17], which combines some groups together to increase per-group sample sizes and allocates a larger fraction of the overall data to separate validation and test sets.

## A.2 Additional experiment details

### A.2.1 Per-group AUROC

Evaluating AUROC within each group measures the strength of the relative ordering of the predictions within that group. This is especially desirable for our experimental purposes should label distributions differ between groups, and is appropriate since we assume groups are known during training and evaluation. We note the limitations of using per-group AUROC in other cases, when assigning per-group classification thresholds is infeasible [5], e.g. when groups are assumed to be unknown.

### A.2.2 Details on §5.2

The pretrained miniature Bert models we use in §5.2 were accessed via `https://tfhub.dev/google/collections/bert/1`. Details on the number of parameters, training time, and pretraining process for each model can be found in [21].

For each fine-tuning run we use the Adam optimizer with learning rate 0.0001 and weight decay of 0.01 and train for 100 epochs with batchsize 64 and 500 gradient steps per epoch. While we fix these parameters as part of the fixed training procedure definition for this experiment (e.g. as relates to definition 1), future work could incorporate hyper-parameter optimizations as part of the training procedures for each (dataset, model) pair.

## A.3 Illustrative example details

Our examples in §4 use the following linear model:
$$y_g = w \cdot x_g + b_A \cdot \mathbb{I}[g = A] + b_B \cdot \mathbb{I}[g = B] + \epsilon_g \quad g \in \{A, B\} = \mathcal{G} \, .$$
Here we give the exact parameters that produce experimental results in fig. 1. In the first example (fig. 1a), we vary the intercept of the true model between the groups, as well as mean of the distribution of the features:
$$w = -1; b_A = -10, b_B = 10 \tag{5}$$
$$x_A \sim \mathcal{N}(10, 5) + \epsilon_A, \quad \epsilon_A \sim \mathcal{N}(0, 1)$$
$$x_B \sim \mathcal{N}(-10, 5) + \epsilon_B, \quad \epsilon_\mathbf{A} \sim \mathcal{N}(0, 1) \, .$$

In the second example (fig. 1b), the same model applies to both groups, but we vary the parameters governing observation noise $\epsilon_g$ between groups, as well as the spread of the feature distribution:
$$w = -1; b_A = b_B = -10 \tag{6}$$
$$x_A \sim \mathcal{N}(-10, 5) + \epsilon_A, \quad \epsilon_A \sim \mathcal{N}(0, 1)$$
$$x_B \sim \mathcal{N}(-10, 2) + \epsilon_B, \quad \epsilon_B \sim \mathcal{N}(0, 10) \, .$$

