# OpenReview forum: "Striving for data-model efficiency: Identifying data externalities on group performance"
_NeurIPS.cc/2022/Workshop/TSRML — TSRML2022_

### Official Review · Reviewer_xQDQ · 2022-10-21

**Overall Rating:** 6

**Summary:**

This paper studied a very interesting problem, data externalities, that adding more data from some input sources reduces performance on key evaluation groups. The authors showed that certain 'data inefficiency' implied space for model improvement, and focused on understanding how and when data externalities manifest across learning settings and training procedures. Extensive real data analysis validate their findings.

**Strengths:**

- Overall, the paper is well written and the motivation is clearly clarified.

- The studied problem is interesting and important for accurate and trustworthy machine learning.

- Extensive experiments are provided and results are impressive, which show the usefulness of the framework.


**Weaknesses:**

- The paper is not easy to follow with too many terminologies which may be benefit from detailed explanations and additional references.

- The authors may consider a figure to illustrate the whole framework of data externalities on group performance.

**Overall Recommendation:**

This paper delivered an interesting idea and useful framework with extensive experiments, but terminologies need more detailed explanations.

**Review Confidence:**

3: The reviewer is fairly confident that the evaluation is correct

---

### Official Review · Reviewer_gQLH · 2022-10-22

**Overall Rating:** 9

**Summary:**

This paper studies the '*data externalities phenomenon*' --- adding more data from certain input sources will reduce performance on key evaluation groups. The authors conduct a set of interesting experiments for understanding the data externalities phenomenon.

**Strengths:**

1. The findings on *data externalities phenomenon* is very interesting, where this paper shows that this phenomenon could occur in both toy and real-world datasets. Meanwhile, the experiments are well-designed and provide valuable insights on understanding this phenomenon (e.g., complicated effect on magnitude of data externalities).

**Weaknesses:**

N/A

**Overall Recommendation:**

I recommend acceptance, because this paper provides interesting results on the data externalities phenomenon.

**Review Confidence:**

4: The reviewer is confident but not absolutely certain that the evaluation is correct

---

### Decision · Program_Chairs · 2022-10-23

**Decision:**

Accept

**Comment:**

Following the unanimous recommendations from reviewers, the submission is accepted.